# Nanosuspension-Based Dissolvable Microneedle Arrays to Enhance Diclofenac Skin Delivery

**DOI:** 10.3390/pharmaceutics15092308

**Published:** 2023-09-13

**Authors:** Luca Casula, Rosa Pireddu, Maria Cristina Cardia, Elena Pini, Donatella Valenti, Michele Schlich, Chiara Sinico, Salvatore Marceddu, Nina Dragićević, Anna Maria Fadda, Francesco Lai

**Affiliations:** 1Dipartimento di Scienze della Vita e dell’Ambiente, Sezione di Scienze del Farmaco, CNBS, Università degli Studi di Cagliari, 09124 Cagliari, Italy; luca.casula@unica.it (L.C.); rosapireddu@unica.it (R.P.); cardiamr@unica.it (M.C.C.); valenti@unica.it (D.V.); michele.schlich@unica.it (M.S.); sinico@unica.it (C.S.); mfadda@unica.it (A.M.F.); 2Department of Pharmaceutical Sciences, General and Organic Chemistry Section “Alessandro Marchesini”, University of Milan, 20133 Milan, Italy; elena.pini@unimi.it; 3Istituto di Scienze delle Produzioni Alimentari (ISPA)—CNR, Sez. di Sassari, 07040 Baldinca, Italy; salvatore.marceddu@cnr.it; 4Department of Pharmacy, Singidunum University, 11107 Belgrade, Serbia; ndragicevic@singidunum.ac.rs

**Keywords:** nanocrystals, nanosuspension, diclofenac, dissolving microneedles, skin delivery, transdermal delivery

## Abstract

Applying a formulation on the skin represents a patient-acceptable and therapeutically effective way to administer drugs locally and systemically. However, the stratum corneum stands as an impermeable barrier that only allows a very limited number of drugs to be distributed in the underlying tissues, limiting the feasibility of this administration route. Microneedle arrays are minimally invasive platforms that allow the delivery of drugs within/across the skin through the temporary mechanical disruption of the stratum corneum. In this work, microneedle arrays were combined with nanosuspensions, a technology for solubility enhancement of water insoluble molecules, for the skin delivery of diclofenac. Nanosuspensions were prepared using a top-down method and loaded in the tips of 500 µm or 800 µm high microneedles. The quality of the combined platform was assessed using electron microscopy and spectroscopic and calorimetry techniques, demonstrating the ability to load high amounts of the hydrophobic drug and the compatibility between excipients. Lastly, the application of nanosuspension-loaded microneedles on the skin in vitro allowed the delivery of diclofenac within and across the stratum corneum, proving the potential of this combination to enhance skin delivery of scarcely soluble drugs.

## 1. Introduction

The skin is considered an attractive site for the painless and non-invasive delivery of therapeutic agents, with the possibility to regulate their release and prevent first-pass metabolism. Following topical absorption, medications can act locally or regionally, at a variety of target sites. Moreover, when using an appropriate formulation, the therapeutic agents can reach the dermis layer—which is rich in blood microcirculation—and be systemically absorbed, this is a process called transdermal drug delivery [1]. Transdermal drug delivery has several advantages, including improved patient acceptability, abolition of gastrointestinal degradation and first-pass metabolism. The main obstacle of this delivery route is represented with the skin barrier function, and in particular the stratum corneum impermeability [2]. For these reasons, various strategies have been investigated to enhance skin permeability, from chemical enhancers to soft matter nanocarriers such as liposomes, which when properly designed could have the capability to overcome these drawbacks [3]. Among the different strategies for improving dermal application, nanocrystals are an extremely intriguing option. The term “nanocrystals” refers to nanoparticles of a pure drug lacking any sort of matrix material and having an average diameter less than 1 µm (usually between 200 and 500 nm), formulated as colloidal suspensions, also known as nanosuspensions (NS), in both water and non-water media and stabilized with either surfactants (ionic or non-ionic) or polymers [4,5,6]. The overall impact of the nanosizing procedure on the improvement of the drug’s skin penetration is related to three factors: (1) an increase in the drug saturation solubility and, as a result, the concentration gradient between the formulation and the skin surface; (2) a faster dissolution process, owing to the increased particle surface area; and (3) a greater nanocrystal adherence to the skin [7]. In our laboratory, we previously optimized the production of a nanocrystal suspension of diclofenac acid (DCF) [8]. DCF is a well-known non-steroidal anti-inflammatory drug (NSAID) characterized with poor aqueous solubility and, similarly to other medications of the same class, with dose-dependent gastrointestinal unwanted effects, including gastric ulcers and the risk of internal bleeding, abdominal pain, nausea and enteropathy [9].DCF nanosuspension was developed with the dual aim of improving its solubility (and bioavailability) [10,11] and reducing the gastrointestinal side effects following the administration through parenteral routes (topical or subcutaneous) [12,13]. When topically applied on the skin, the DCF nanocrystal formulation exhibited higher skin accumulation compared to the raw drug [8]. In an effort to enhance the effectiveness of the nanocrystal technology, DCF-NS were also combined with other devices to boost their delivery through the skin, such as microneedle rollers and a needle-free jet injector, resulting in the formation of drug nanocrystal depots [14,15]. A lot of interest has recently arisen around polymeric microneedles arrays (MNAs) as an alternative technology to enhance drug delivery and distribution through the skin [16]. Microneedle arrays can be described as small devices with microscopic needle-like projections that can pierce the stratum corneum and form channels that allow macromolecules and nanoparticles to cross the skin barrier more effectively. They can vary in length (25–2000 µm) and geometry, and may be made from a variety of materials (silicon, ceramics, glass, sugars, biodegradable polymers, steel, etc.) [17,18]. More specifically, dissolving MNAs are prepared using water-soluble and biodegradable polymers that dissolve after insertion and release the active compound (poke and release mechanism), with the possibility of modulating the release kinetics by controlling the dissolution rate of the MNAs’ matrix [19,20]. However, this particular class of MNAs usually shows a limited drug loading capability, especially with hydrophobic drugs [21,22]. To overcome this issue, the nanocrystal technology can be exploited to obtain homogeneous polymeric dispersion with high concentration of poorly water-soluble drugs for the preparation of MNAs [23,24]. In this study, a DCF-NS was prepared with a wet ball media milling technique and then used to prepare Polyvinylpyrrolidone (PVP)-based dissolving MNAs (DCF-NS-MNAs), with needle lengths of 500 and 800 µm. Both DCF-NSs and DCF-NS-MNAs were characterized in terms of morphology using scanning electron microscopy (SEM) and solid-state characterization was carried out using differential scanning calorimetry (DSC) and Fourier transform infrared spectroscopy (FT-IR) to evaluate the compatibility among the different component of DCF-NS-MNs. Finally, in vitro penetration/permeation studies were conducted in porcine skin using the Franz vertical diffusion cells.

## 2. Materials and Methods

### 2.1. Materials

Diclofenac sodium salt was purchased from Galeno (Comeana, Italy). Poloxamer 188 (P188) and Polyvinylpyrrolidone (MW 10 kDa) were purchased from Sigma–Aldrich (Milan, Italy). MPatch™ Microneedle Templates ST-14 (10 × 10 array, H = 500 μm, base = 200 μm, pyramid) and ST-29 (10 × 10 array, H = 800 μm, base = 200 μm, pyramid) were obtained from Micropoint Technologies Pte Ltd., Singapore. Water was purified with a Milli-Q system with a 0.22 µm Millipak 40 filter (Millipore, Dublin, Ireland). All other reagents and solvents were of analytical grade.

### 2.2. Preparation of DCF-NS

Following a published procedure for the production of a crystalline solid, the acid form of diclofenac was prepared from its sodium salt [8]. Concisely, diluted hydrochloric acid was added to a saturated water solution of diclofenac sodium salt to form a white precipitate of diclofenac acid, until no additional precipitate was formed. The precipitated solid was then filtered, washed thoroughly with distilled water, air dried and used for the nanosuspension preparation. The formulation was prepared through a wet ball media milling technique, using a 2:1 (*w*/*w*) drug/stabilizer ratio. The bulk drug (10 mg/mL) was weighted and dispersed in an aqueous solution of P188 (5 mg/mL), using an Ultra Turrax T25 basic for 5 min at 6500 rpm. The obtained suspension was transferred in 1.5 mL conical tubes containing approximately 0.4 g of 0.1–0.2 mm yttrium-stabilized zirconia–silica beads (Silibeads^®^ Typ ZY Sigmund Lindner, Warmensteinach, Germany), and oscillated at 3000 rpm for 70 min using a bead-milling cell disruptor device (Disruptor Genie^®^, Scientific Industries, Bohemia, NY, USA). The obtained nanosuspension was then separated from the milling beads by sieving.

### 2.3. Particle Size Analysis

Dynamic Light Scattering (DLS) was used to determine the average diameter and polydispersity index (PDI, as a measure of the size distribution width) of the samples with a Zetasizer nano (Malvern Instrument, Worcestershire, UK). Samples were backscattered using a helium–neon laser (633 nm) at an angle of 173° and a constant temperature of 25 °C. The Zetasizer nano was also used to estimate the Zeta potential value by means of the M3-PALS (Phase Analysis Light Scattering) technique. Just before the analysis, nanosuspensions were diluted with distilled water. All the measurements were made in triplicate.

### 2.4. Preparation of DCF-NS/PVP Dispersion and PVP Solution

DCF-NS/PVP dispersion was prepared by adding 0.8 g of PVP to 1 mL of DCF-NS and gently stirring overnight to obtain a homogeneous sample. Similarly, the PVP solution was obtained following the same protocol, using water instead of DCF-NS.

### 2.5. Preparation of DCF-NS-MNAs

MNAs were prepared with the solvent casting technique (Figure 1). Briefly, 60 µL of DCF-NS/PVP dispersion were homogenously added in the microneedle template for the first deposition. The template was then carefully placed in a 50 mL falcon tube and centrifuged using a NEYA 8 BASIC centrifuge (Neya Centrifuges, Carpi, Italy) at 4000 rpm for 15 min to fill the needles part of the array and eliminate the air bubbles. After the centrifugation, MNA was left to dry overnight. Additional cycles of deposition–centrifugation–drying were carried out by adding 50 µL of the PVP solution to completely fill the template and form the base of MNAs. After the last deposition, MNAs were left to dry at 25 °C for 3–5 days in a glass desiccator jar protected from light. The arrays were carefully extracted from the templates by placing a little piece of double-sided tape and gently pulling out. MNAs were then vacuum packed until use.

### 2.6. Preparation of DCF-NS-Disks

DCF-NS-Disks were prepared to obtain a solid PVP system with a flat surface (without the invasive needle-like projections) and were loaded with the same DCF-NS amount of DCF-NS-MNAs. Briefly, 50 µL of DCF-NS/PVP dispersion was pipetted on a flat support, let dry overnight and then vacuum packed until use.

### 2.7. Scanning Electron Microscopy

Scanning electron microscopy (SEM) was used to evaluate the morphology of DCF-NS. On a silicon wafer, a 3 µL drop of DCF-NS was applied and allowed to air dry. The sample was imaged using a SEM (Jeol, Japan) operating at 10 kV after being sputter coated with a 10 nm layer of gold. DCF-NS-MNAs were imaged using a Zeiss ESEM EVO LS 10 (Oberkochen, Germany) at variable pressure (VP). Without any kind of pre-processing, MNAs samples were mounted onto aluminum stubs and imaged while running at 20 kV in VP.

### 2.8. DCF Quantification in MNAs

To evaluate the amount of DCF effectively incorporated in the needles, DCF-NS-MNAs were dissolved in 2 mL of MeOH, appropriately diluted and analyzed via HPLC for drug quantification. The loading efficiency (LE%) has been calculated as the ratio between the amount of DCF detected with HPLC and the amount of DCF used to prepare MNAs.

### 2.9. HPLC Analysis

Using a high-pressure liquid chromatograph (HPLC), Alliance 2690 (Waters Corp, Milford, MA, USA), equipped with a photodiode array detector and a computer integrating apparatus (Empower 3), the quantitative determination of DCF was carried out. A Sunfire C18 column (3.5 µm, 4.6 × 150 mm, Waters) was used for the analyses. The mobile phase contained acetonitrile, water and acetic acid (59.225:40.75:0.025 *v*/*v*), and it was eluted at a flow rate of 0.5 mL/min. Using an auto sampler, samples (10 µL) were injected, and DCF was detected at 280 nm. Using standard solutions obtained by diluting the stock standard solution with the mobile phase, a standard calibration curve (peak area of DCF vs. known drug concentration) was created. Calibration graphs were plotted according to the linear regression analysis, which gave a correlation coefficient value (R^2^) of 0.999. The limit of detection was 1 ng, while the limit of quantification was 2 ng. Sample preparation and analyses were performed at room temperature.

### 2.10. Solid State Characterization

To evaluate any possible interactions among the used substances, or modifications due to the production of the NS or of MNAs, solid state analyses were carried out singularly on the different components of DCF-NS-MNAs (DCF, P188, PVP) and on their physical mixtures (PM) with a 1:1 ratio (DCF + P188, PVP + DCF, PVP + P188) or a 1:1:1 ratio (PVP + DCF + P188). Moreover, DCF-NS alone and its physical mixture with the polymer were also analyzed (DCF-NS+PVP). To prepare PMs, the components were weighed, mixed and uniformly ground with a mortar and pestle. Finally, empty Disks and DCF-NS-Disks were also investigated to evaluate the final system composition. Since the preparation process and the composition of Disks were identical to MNAs, they were used instead of the latter for a mere operative reason.

FT-IR spectra over the range of 4000–650 cm^−1^ at a resolution of 4 cm^−1^ were acquired with a Perkin Elmer Spectrum One FT-IR (Perkin Elmer, Waltham, MA, USA), and equipped with a Perkin Elmer Universal ATR sampling accessory. The samples were deposited onto a diamond crystal plate.

Differential Scanning Calorimetry studies were carried out using a Perkin Elmer DSC 4000 (Perkin Elmer, Waltham, MA, USA). Sample weights of 2–3 mg were sealed in aluminum pans; the heating rate was kept at 10 °C/min from 30 ° to 250 °C purging nitrogen at a flow rate of 20 mL/min. The DSC instrument was calibrated with the melting temperature of indium (156.6 °C).

### 2.11. In Vitro Skin Permeation of DCF-NS with MNAs

The ability of MNAs to enhance DCF-NS penetration and/or permeation through the skin was evaluated using full-thickness skin of newborn Goland–Pietrain hybrid pigs (1–1.5 kg), which died of natural causes and were provided by a local slaughterhouse. The experiments were performed with the Franz vertical cells model, with a diffusion area of 0.785 cm^2^ and DCF-NS-Disks were used as control. The skin was excised from the animal and stored at −80 °C until the day of the experiment. Skin specimens (*n* = 6 per formulation) were pre-equilibrated with saline (NaCl 0.9% *w*/*v*) at 25 °C. The excess of saline on the skin was removed by gently pressing paper towel on it and MNAs were applied on the stratum corneum. The specimens were then placed above the receptor compartments, which were filled with 5.5 mL of saline solution (NaCl 0.9% *w*/*v*) and kept stirred and thermostatted at 37 ± 1 °C for 8 h. After 8 h, the receiving solution was entirely withdrawn, freeze dried, re-dispersed with methanol and vortexed for 30 s to extract the drug. The skin surface was washed with 1 mL of distilled water and dried with filter paper. Tesa^®^ AG (Hamburg, Germany) adhesive tape was used to strip off the stratum corneum. Each piece of adhesive tape was firmly applied to the skin’s surface before being swiftly removed in one motion. Using a surgical scalpel, the epidermis and dermis were separated. To extract the accumulated drug, skin strata were cut, each stratum was placed in a flask with methanol and was sonicated for two minutes. The tape, the tissue suspensions and the methanolic dispersion of the receptor compartment were filtered and assayed for drug content with HPLC.

### 2.12. Statistical Analysis

Results are expressed as the mean ± standard deviation. Multiple comparisons of means (one-way ANOVA with post hoc Tukey HSD test) were used to substantiate statistical differences between groups, while Student’s *t*-test was used to compare two samples. Data analysis was carried out with the software package XL Statistic (XLSTAT 22.5) for Excel (Microsoft, Redmond, WA, USA). Significance was tested at the 0.05 level of probability (p).

## 3. Results and Discussion

### 3.1. Preparation and Characterization of Nanosuspension

Drug nanocrystals were prepared using a wet ball media milling technique in order to obtain a homogeneous water-based nanosuspension. DCF concentration was set at 1% (*w*/*w*), and 0.5% (*w*/*w*) Poloxamer 188 was used as a non-toxic stabilizer.

As shown in Table 1, examination of colloidal properties with DLS showed an average nanocrystal diameter of around 230 nm, with a polydispersity index (PDI) of 0.16, indicating a narrow size distribution. As for Zeta potential measurements, the sample revealed a substantially negative value (−38 mV), indicative of good nanosuspension stability. Characterization data are in agreement with our previous study, where the stability of the system was also proved with DLS analysis after 90 days of storage [8]. As a part of the quality control of the production, the morphology of DCF-NS was also assessed using SEM, that allowed us to evaluate the shape of the crystals and the possible variations induced by the nanosizing process.

DCF raw material appears as irregularly thin and elongated crystals (Figure 2a). Conversely, nanosizing through the wet media milling method led to the formation of nanocrystals with regular and a more rounded shape (Figure 2b). Microscopy analysis also confirmed the homogenous particle size distribution evidenced with DLS.

### 3.2. Preparation and Characterization of DCF-NS-MNAs

After the characterization of DCF nanosuspension, dissolving PVP microneedles were prepared using the solvent casting technique. The process consisted of successive cycles of deposition of the polymeric solution–centrifugation–drying in a desiccator. The first deposition was carried out using the dispersion of DCF-NS in PVP (DCF-NS/PVP) to fill the needle tips with high drug amounts. For the following depositions, aimed at forming the base of MNAs, a solution of PVP was used.

As can be seen in the macroscopic images (Figure 3a,b), MNAs were formed correctly and 100 microneedles arranged in a 10 × 10 matrix were obtained from each mold, with the shape of each microneedle clearly visible. To further investigate their structure, the samples were visualized using SEM under variable pressure conditions. Both the samples with the needle’s length of 500 and 800 µm (Figure 3c,d, respectively) showed a linear disposition of the needles, with a regular pyramidal shape and sharp extremities. 

To evaluate the efficacy of the production process, MNAs were also characterized in terms of drug loading. As previously detailed, DCF-NSs were dispersed in a PVP solution, and such mixture was employed to fill the microneedles tips of the silicon mold. For each MNA, 600 µg of nanosized DCFs were deposited on the molds. To evaluate the actual drug loading, MNAs were dissolved in methanol, and the resulting solution was analyzed with HPLC. Results revealed a DCF loading of 554 ± 68 µg for the 500 µm DCF-NS-MNAs and 543 ± 33 µg for the 800 µm ones, resulting in a loading efficiency of 92 ± 11% and 90 ± 5%, respectively. These values suggest that only a minimal part of the drug was lost, and are in line with a previous study, where an iron-chelating insoluble molecule was loaded in microneedles as a nanosuspension [21].

### 3.3. Solid State Characterization

#### 3.3.1. Characterization of DCF-NS

Polymorphic changes of nanosized drugs and drug-excipient compatibility can be studied using several thermal and non-thermal methods. At first, FTIR and DSC analyses of DCF; P188, their physical mixtures (PM) in a 1:1 ratio (*w*/*w*); and DCF-NS were performed.

As concerns DCFs, two monocline forms, HD1 and HD2, are reported in the literature [25]. As shown in Figure 4a, the ATR spectrum of DCFs showed an intense peak at 3322 cm^−1^ due to the N-H stretching typical of the of HD2 form of diclofenac [14], the peak at 1690 cm^−1^ corresponded to the C=O stretching, the COO stretchings at 1577 and 1452 cm^−1^ and C=C and C-O bands at 1506 and at 1159 cm^−1^, respectively. The P188 spectrum, in agreement with the literature’s data [26], exhibited in the spectral range between 3600 and 3400 cm^−1^ of the OH stretching; at 2969, 2882 and 2740 cm^−1^ of the C-H sp^3^ stretching bands and at 1107 cm^−1^ of the C-O stretching. The ATR analyses of the PM and DCF-NS are the superimposition of components spectra, indicating the full compatibility of the drug with the stabilizer used in the formulation. In both samples, the presence of the N-H peak at 3322 cm^−1^ suggests that the original polymorphic form of DCF is retained following the NS fabrication process.

As shown in Figure 4b, the DSC thermogram of P188 exhibited a sharp peak at 56 °C with a ΔH = 113.7762 J/g. The thermal behavior of DCF acid showed a sharp endothermic peak at 176 °C corresponding to its melting point, with an enthalpy of 129.44 J/g, indicating its crystalline nature; this peak disappeared in the DSC thermogram of PM, suggesting a dissolution of DCF in P188 with the increasing of temperature. In the lyophilized NS thermal analysis, the two endotherms of the components are depicted with a slight decrease in melting points, 49 ° and 159 °C, respectively; the decrease of DCF enthalpy at 42.32 J/g may be attributed to a loss of systemic order due to the formulation process.

#### 3.3.2. Characterization of DCF-NS Disks

In order to study DCF stability and the interactions with the excipient used for the fabrication of MNAs, PVP, physical mixtures of P188/PVP, DCF/PVP, DCF-NS/PVP (1:1 *w*/*w* ratio), DCF/P188/PVP (1:1:1 *w*/*w* ratio), empty Disks and DCF-NS-Disks were also analyzed. In these experiments, Disks were employed as a substitute of MNAs for their faster fabrication and easier manipulation. We considered Disks to be a suitable model to assess the interactions between the components, as their composition and method of desiccation were identical to those employed for the fabrication of MNAs.

The FTIR spectrum of PVP K10 (Appendix A) displayed a broad peak at 3412 cm^−1^ due to O-H stretching vibrations of absorbed water, at 2953, 2916 and 2886 cm^−1^ of the Csp^3^– H stretchings, at 1646 cm^−1^ of the C=O band, between 1493 and 1422 cm^−1^ of the C–H deformations and at 1287 of the N–C stretching.

All the spectra of PM (Appendix A) evidenced typical bands of the components without any new peaks suggesting no interaction between them.

The ATR spectra of empty and DCF-NS-loaded Disks (Figure 4a) are superimposable; in fact, only peaks related to PVP are visible considering the low DCF amount in the final formulation. The absence of DCF and P188 bands could be due to the low DCF percentage in the formulation, or to the incorporation of DCF-NS into the formulation matrix.

The thermal behavior of PVP K10 (Appendix A), in agreement with the reported data [27], is typical of a hygroscopic and amorphous excipient, with a wide endothermal event in the 30–120 °C range due to polymer dehydration and a weak glass transition at about 144 °C. The physical mixture of the polymers, P188 and PVP (Appendix A), showed similar thermal profiles as the individual polymers, without the appearance of new thermal peaks. DCF/PVP analysis showed the dehydration endotherm of PVP and the melting event of DCF at 168 °C with a ΔH = 9.2483 J/g (Appendix A), while DCF/P188/PVP thermogram (Appendix A) presented the P188 endotherm, and the endothermic event related to the drug melting was significantly reduced.

The DCF-NS/PVP physical mixture showed the sharp endothermic event of P188 and the loss of water of PVP and a weak peak of DCF at 156 °C, with a ΔH = 1.0564 J/g (Appendix A).

As depicted in Figure 4b, the DCF-NS-loaded Disks exhibited a sharp endothermic peak at 122 °C (ΔH = 128.4831 J/g) absent in the empty drops, probably due to the release of DCF-NS from the polymeric matrix.

### 3.4. In Vitro Skin Permeation Studies

The positive results in terms of quality of the DCF-NS-MNAs system prompted us to perform in vitro penetration and permeation experiments using Franz diffusion cells and newborn pig skin. In this study, dermal and transdermal delivery of DCF was evaluated using DCF-NS-MNAs with 500 µm or 800 µm needle lengths. The DCF-NS-Disks were also applied on the skin and used as a reference for a device with identical composition without the invasive needle part.

As reported in Figure 5, it can be immediately noted that needle height positively correlates with the amount of drug recovered in the receptor compartment after 8 h of application. When 800 µm needle MNAs are used, 2.9 µg/cm^2^ DCFs successfully cross all the skin strata and are recovered in the receptor compartment, which simulates the systemic absorption. This value is reduced to 1.4 µg/cm^2^ DCF when the needle length is 500 µm and drops to zero when the skin is left intact by the application of needle-less DCF-NS-Disks. Needle length has also an impact on the amount of drug accumulated in the stratum corneum, which is almost 5 times higher when the higher MNAs are employed (217 µg/cm^2^), compared to the shorter ones (44 µg/cm^2^). This difference could be explained by the amount of drug nanocrystals loaded in the needles and the protocol used for the preparation of MNAs. In fact, even though the same amount of DCF-NS/PVP dispersion is used for the preparation of the two types of MNAs (500 µm and 800 µm), the structure of the two molds lead to a different deposition of the drug in the array/patch. In case of the 500 µm samples, the DCF-NS/PVP dispersion fills the needles, and the excess creates a thin layer above them. This portion of the drug will not be able to deposit in the skin’s layers, as it is located in the non-invasive part of the array. In contrast, the needle tips in the 800 µm molds can host the entire volume of DCF-NS/PVP dispersion, with few or no excess left to form the above-mentioned layer. Therefore, when DCF-NS-MNAs 800 µm are applied on the skin, a higher amount of the loaded drug can permeate in the skin’s layers since it is more efficiently located in the needle tips.

In sum, the remarkable ability of microneedles to enhance transdermal drug permeation was demonstrated, holding great promises for evolving drug administration with a minimally invasive and effective method, with the potential to boost therapeutic outcomes and patient compliance.

## 4. Conclusions

As widely reported, microneedle-mediated drug delivery systems provide high acceptability and safety due to many factors such as reduced pain, risk of transmission infection and needle stick injuries, as well as ease of self-administration and enhanced acceptance in children administration. In this work, we combined two emerging technologies—nanosuspensions and microneedle arrays to promote skin delivery of the anti-inflammatory diclofenac. The thorough characterization of each part of the system and the spectroscopic/calorimetric studies of binary–ternary mixtures of drug and excipients proved the quality of the production process and the chemical compatibility between components. Lastly, in vitro pig skin permeation studies provided a proof of concept of the ability of microneedle arrays to promote the accumulation of diclofenac within and across the skin in a needle-length dependent fashion. Overall, this work opens new avenues for the efficient delivery of insoluble anti-inflammatory molecules across the skin with a minimally invasive system and provides the basis for the development of combination products and related mechanistic studies on skin delivery. Further in vivo investigation is needed to demonstrate safety, efficacy and reliability of this promising approach.

## Figures and Tables

**Figure 1 pharmaceutics-15-02308-f001:**
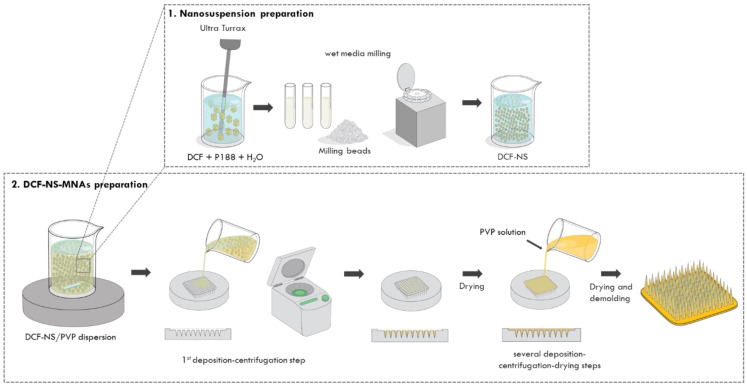
Schematic representation of the preparation of DCF-NS and DCF-NS-MNAs.

**Figure 2 pharmaceutics-15-02308-f002:**
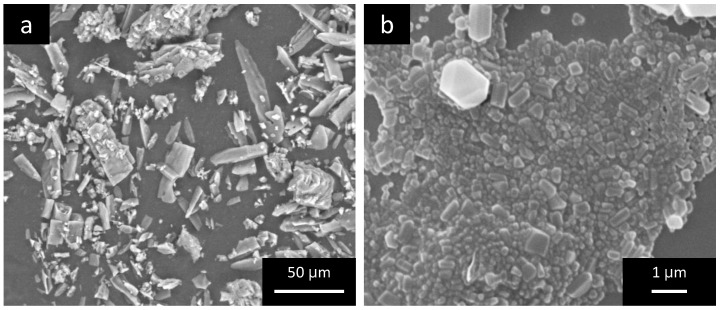
SEM images of DCF raw material (**a**) and DCF nanocrystals (**b**).

**Figure 3 pharmaceutics-15-02308-f003:**
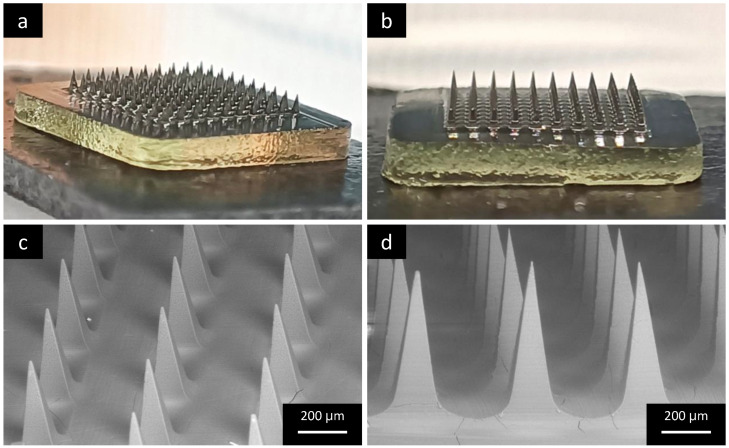
(**a**,**b**) Representative macroscopic images of DCF-NS-MNAs (needles length 800 µm) and SEM images of DCF-NS-MNAs with needle lengths of (**c**) 500 µm and (**d**) 800 µm.

**Figure 4 pharmaceutics-15-02308-f004:**
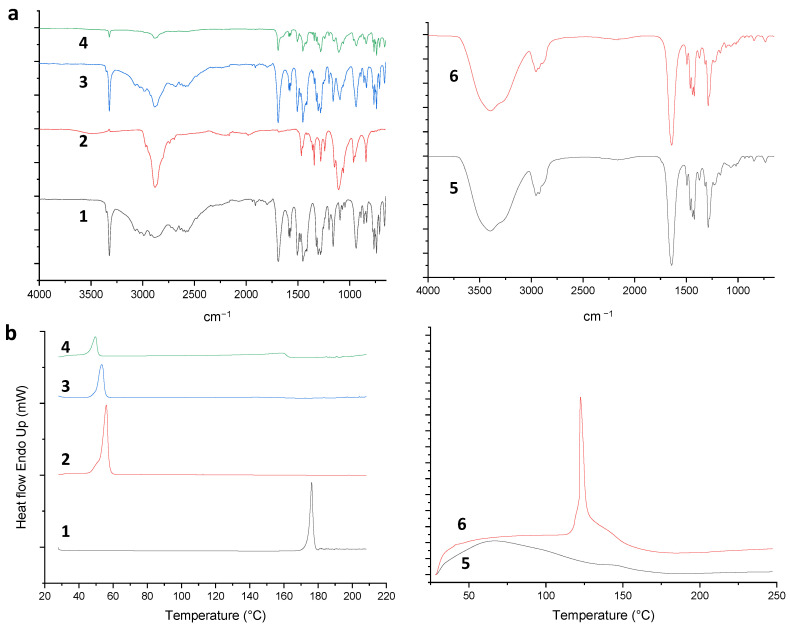
ATR-FTIR spectra (**a**) and DSC thermograms (**b**) of (1) DCF, (2) P188, (3) DCF/P188 PM, (4) DCF-NS, (5) empty Disks and (6) DCF-NS-Disks. DSC thermogram of (5) empty Disks and (6) DCF-NS-Disks.

**Figure 5 pharmaceutics-15-02308-f005:**
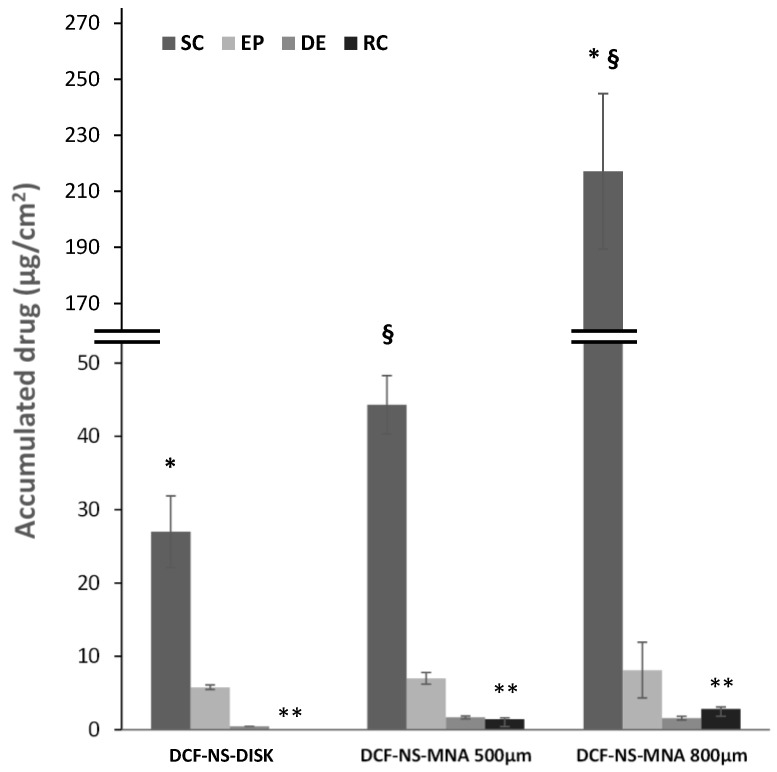
Cumulative amount of DCF retained into and permeated through newborn pig skin layers after 8 h application of microneedle arrays loaded with DCF-NS with needle length of 500 µm (DCF-NS-MNAs 500 µm), 800 µm (DCF-NS-MNAs 800 µm) or DCF-NS-Disks. SC—stratum corneum; EP—epidermis; D—dermis; RC—receptor compartment. Same symbols indicate couple of values that are statistically different (*p* < 0.01).

**Table 1 pharmaceutics-15-02308-t001:** Composition and characterization of DCF nanosuspension at the day of preparation.

DCF Nanosuspension Composition		Characterization		
Component	% (*w*/*w*)	Mean Diameter (nm)	PDI	Zeta Potential (mV)
DCF	1.0	230 ± 4	0.16 ± 0.02	−38 ± 1
Poloxamer 188	0.5			
Water	98.5			

## Data Availability

Data are available from the authors upon reasonable request.

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
