# Peer review of "Nanosuspension-Based Dissolvable Microneedle Arrays to Enhance Diclofenac Skin Delivery"

_pharmaceutics, 2023, doi:10.3390/pharmaceutics15092308_

Round 1
Reviewer 1 Report
Reviewers Comments1#
Comments for the manuscripts entitled “Nanosuspension-based dissolvable microneedle arrays to 2 enhance diclofenac skin delivery The focus of the study was on microneedle arrays were combined with nanosuspensions, a technology for solubility enhancement of water insoluble molecules, for the skin delivery of diclofenac. I would like to address some queries that need to be rectified before acceptance. While the researchers have conducted commendable experimental work, this manuscript is technically suitable for publication One more drawback of this paper is less than acceptable is that the characterization of prepared nanocomposites, binding phenomena, and less variability of SEM images. Introduction feels strange to read and should require some re-writing (see comments to author). Images have have errors in them and need fixing. The whole article would benefit from simple 'spell-checker'. Meanwhile, the target is interesting and the developed approach performed well. Before the manuscript could be accepted for publishing, there are a few questions need to be addressed
1. This paper looks like a copy of a comprehensive thesis. It needs a lot editing. The methodology section contains unwanted details while relevant information are missing. Methodology needs to be logically arranged.
2. The Introduction is a very long section with a lot of unwanted information. Authors should confine to one page giving important facts and justification of the work. Results and discussion section contains a lot of unwanted information and works of other authors. The results and discussion should have only the important results and the discussion should focus on the results. Remove all the unwanted information.
3. In the FTIR spectrum, (Fig 4) Author should mention the important peak value in the shown spectrum
4. Kindly make sure that correct scientific nomenclature is used throughout the manuscript (e. g., units, Latin names for biopolymers).
5. Review the symbol of 0C, K, minutes, ml and hour in whole manuscript
6. Fig. 4 the FT-IR data cannot give solid evidence for the formation of DCF/P188 PM, (4) DCF-NS, (5) EMPTY Disks and (6) DCF-NS-Disks. since there is almost no significant difference between curves 1,2,3,4,and 5, 6. X axis of the FTIR is should be cm-1. It would be better to add XPS characterization.
7. What is the size (crystalline size and particle size) and shape of DCF/P188 PM?
8. Biocompatibility is one of the major factor for in vivo and vitro analysis, author should also address this impotent issue
9. Conclusion section should be more targeted & presented pointwise along with future prospective
10. throughout the manuscript: separate the numbers from the units (i.e.: 5 mg instead of 5mg)
11. Review the symbol of 0C, K, minutes, ml and hour in whole manuscript
12. DSC curve should be logically explained along with temperature and anount of heat and correspond temperature
13. I would like to recommend to add the TGA analysis and BET analysis for pore size and surface volume of study of prepared nanosuspension. .
The paper should be minor revision and complete characterization and explanation of the prepared materials is in order before this work can be further considered for publication.
Minor editing of English language required
Reviewer 2 Report
The manuscript titled: “Nanosuspension-based dissolvable microneedle arrays to 2 enhance diclofenac skin delivery” by Luca Casula et al. presents an innovative approach to improve transdermal drug delivery using a combination of nanosuspensions and microneedle arrays. The introduction effectively establishes the significance of the research by highlighting the limitations of conventional drug delivery methods and introducing the potential benefits of nanosuspensions and microneedle arrays. The materials and methods section provides a detailed description of the experimental procedures, ensuring reproducibility. The results section presents thorough characterization data for the nanosuspension and microneedle arrays, supported by well-constructed figures and tables. The discussion successfully links the findings to the broader context of transdermal drug delivery and highlights the potential impact of the study.
In my opinion, this manuscript will enhance interest of the scientific community of the Pharmaceutics journal and fits well to its scope. The article is clear, well organized, and well written and overall, I recommend this work for publication however some concerns below should be addressed:
1. Study is lacking in vivo validation which restricts the assessment to in vitro conditions.
2, The study's focus solely on diclofenac limits its generalizability to other drugs.
3. While the stability of the nanosuspension is briefly mentioned, a more comprehensive discussion on long-term stability and potential degradation would add value.
4. Discussing clinical considerations, such as safety concerns and patient comfort associated with microneedle arrays, would provide a more holistic view of the proposed approach.
Acceptable
